# The Puzzle of Revenge

**Bill Powers**

Independent Researcher, White, SD 57276, USA; wjp@swcp.com

**Abstract:** We pursue a multi-leveled phenomenological exploration of revenge. Revenge's puzzle is to give an account of what exactly revenge accomplishes when it apparently cannot alter the past or remedy the initiating harm. The structure of revenge consists of one harmed, the perception of harm and suffering, and one perceived as responsible for the harm. The situation is apperceived as a negatively saturated experience; as such, it binds and has a hold on the one harmed, constituting her as enthralled. Revenge seeks to remedy the situation by the intentional act of objectifying, constituting, and finitizing the infinite situation. This is accomplished by constituting the guilty one as guilty, thereby mastering, in some measure, the saturated situation. We suggest that the realm and machinery required for this process is found in the realm of the imagination, where similarity and association of ideas and concepts are at play. Saturation plays at the edge of this realm as alien. It is by way of the familiar and constituted that the alien is tamed, and revenge puts the situation to "rest".

**Keywords:** phenomenology; revenge; saturation

## 1. The Puzzle of Revenge

In revenge we are presented with a profound puzzle. Necessary for revenge is the assignment of the guilty, one that we associate with the harm done to us. To revenge this harm, we seek to harm the guilty one and do them violence. Othello killed Desdemona, the woman he profoundly loved, out of revenge. How exactly does the revenge address the harm done? What is the seemingly necessary link between the harm done to you and the harm you inflict on another?

You are harmed. Let us not say how. Let us also say that we attribute the harm to some cause, something that is immediate, something like a direct cause, one that can be seen, and is phenomenally close, perhaps importantly temporally close, to the harm. What is going on when we seek revenge on this other, this other that we hold "responsible", meaning that it is their "fault"?

The apparent notion behind revenge is "tit for tat". This rule is at work in more than human interactions. When someone bangs on a computer keyboard, or destroys a board that is frustrating their efforts to nail it into another board, this appears to be an application of the same rule. Some might object that this is merely frustration, and when we are frustrated we are acting differently than when someone murders our child. Clearly the murder of a child and a resistant object are different, and yet in both we respond to something that we do not want.

Perhaps the frustrated act of destruction shows us what is at stake. When we witness someone smashing their car or some project they were working on, we cannot help but think how senseless it is. After all, we might say, "It's not the car's fault that it won't start". Such senses imply that were it the car's fault, it would make sense to take a tire iron to it. Is it really "fault" that changes everything? If the car will not start, you have to get somewhere soon, and you have exhausted all your expertise to discover and correct the problem, does it make sense to smash the hood and break the windows? That depends on what we mean by "sense". If we regard the car as an inanimate object, and if our understanding of what

keeps a car from starting is in any way correct, then it makes no sense to smash the hood and windows.

Were we to stop there, all we would conclude is that the person smashing the car is acting "irrationally". But our aim here is to make sense of this violent behavior. This tells us that our assessment that he is acting in a senseless manner is itself senseless. We must look elsewhere to find the sense. Suppose the car was Balaam's donkey. Would it make sense then to break it and yell at it? It would certainly make more sense than doing so to an inanimate object. However, if we are to regard the car as somehow animate, it seems that we have left behind the notion that the car is an inanimate machine, for which schematics and rigid mechanical rules make sense. We have re-envisioned the car as something more akin to a living organism, indeed, more like us. Our understanding of the car now becomes a reflection of our understanding of ourselves. We know what it is like to resist, and we have some idea of what it would take to get us to stop resisting.

Is it possible that the intimate Heideggerian relationship of being-ready-to-hand, wherein we become united to the object, provides or discloses the possibility of a relationship whereby the object and we are so united that it takes on our attributes? Were this the case, then it would seem that such frustration and such "revenge" would only occur where we are in some important way united with the object. This would happen in our projects. Where we do not care what time we arrive or whether we arrive or not, it does seem that we would not get frustrated and not take it out on the car. If this makes sense, then an important part of this frustration and "revenge" is the "unity" with an object, another person. We are caught up and entangled with this other, broadly construed. That other can, of course, be our selves. Is there then a confusion of persons? If we regard the car as an important part of our lives, as something reliable, and trustworthy, then this not starting is beginning to look like some kind of betrayal.

Is it, then, a prerequisite to revenge that some kind of a relationship is broken? This is easier to see with the ready-to-hand. The ready-to-hand becomes not-ready-to-hand. We experience the brokenness, which is different from merely something that does not work. It is the relationship that is disclosed as broken. Without the prior relationship, there is no brokenness. The relationship of ready-to-handedness is not merely a mechanical one, one in which cars will not start because of a solenoid. This is because the relationship of ready-to-handedness is not a mechanical relationship. It is a human one. What is bound up in this relationship? Surely a kind of trust, one in which the object can be brought close, even gladly.

So, have we learned anything from this examination of the senseless smashing? I want to argue that the smashing is about a human relationship. In the breaking, Dasein falls into a kind of Angst, perhaps like that of Being-towards-death. After all, there is a limited and small sense in which the world falls away with the breaking of the intimate and trusting relationship with the world and Dasein's place there. It is a rupture, an interruption in the *heimlichkeit* of the world. But why respond this way: with violent vengeance? Even if it is experienced as a kind of betrayal, why seek vengeance?

Why when someone (or thing) betrays us do we seek vengeance? What do we think we are doing? We say, "getting even". But why "get even"? What is being evened? And why should it matter that it is evened? What we want is to have things restored. We want that it had never happened. We cannot exactly do that. So are we doing the best that we can?

Max Scheler, in his book *Ressentiment*, says:

> *[revenge] must be clearly distinguished from the impulse for reprisals or self-defense, even when this reaction is accompanied by anger, fury, or indignation. If an animal bites its attacker, this cannot be called "revenge". Nor does an immediate reprisal against a box on the ear fall under this heading. Revenge is distinguished by two essential characteristics. First of all, the immediate reactive impulse, with the accompanying emotions of anger and rage, is* **temporarily or at least momentarily checked and restrained***, and* **the response is consequently postponed to a later time** *and to a more suitable occasion*

*("just wait till next time"). This blockage is caused by the reflection that an immediate reaction would lead to defeat, and by a concomitant pronounced feeling of "inability" and "impotence". Thus even revenge as such, based as it is upon an experience of impotence, is always primarily a matter of those who are "weak" in some respect. Furthermore, it is of the essence of revenge that it always contains the consciousness of "tit for tat", so that it is never a mere emotional reaction [bold added for emphasis].* (Scheler 2015, p. 5)[1]

Following Scheler, we would say of revenge that it is associated with harm for which we hold someone (something) responsible, and seek to harm as a result, but that the accomplishment of that harm is importantly delayed. Revenge is then to be distinguished from the frustration previously discussed, even while it may have important similarities. We might say that revenge is a delayed act of frustration. In this frustration, where we might take a tire iron to a car, we feel a similar kind of impotence that is found in revenge. In the former case, we take out our "revenge" immediately, while in the latter it is delayed. So it is not the impotence that accounts for the difference. Scheler wants to distinguish the response to a "box on the ear" from revenge, and we will do likewise. If someone slaps you on the face, there appears to be an immediate reflex to strike back. But this response will be mediated by who has slapped you and the context in which it is done. If that someone is a person in authority and perceived as being in a superior position, we may very well not strike back, despite our desire to do so. In this case, our response is delayed.

Scheler nuances this understanding by arguing that revenge will not take place as long as the person injured accepts their inferior position. He says,

*A slave who has a slavish nature and accepts his status does not desire revenge when he is injured by his master; nor does a servile servant who is reprimanded or a child that is slapped. Conversely, feelings of revenge are favored by strong pretensions which remain concealed, or by great pride coupled with an inadequate social position.* (Scheler 2015, p. 7)

We can add, then, to our taxonomy of revenge that the one seeking revenge must not only feel temporarily impotent, but must believe in some sense that they are equals, or can become potentially equals, to the one that has harmed them. They place in abeyance their intended harm for a future opportunity, something they would not do if they accepted this relationship to be significantly "inferior" or relative impotence insurmountable. Revenge is, then, a temporally extended intentional object.

Nonetheless, revenge still bears many similarities to acts of frustration. Acts of frustration require some kind of intimate relationship with that on which we take out our "revenge". An insult is only possible when we are in a position where it can strike home. And this requires a perceived social context within which we are in some sense caught up and intertwined so that something is at risk, and we are exposed in order for a harm to be perceived. Perhaps in the frustrated act, as in taking a tire iron to our car, we feel "equal" to the car; and while feeling impotent and blaming the car, in our "equality" or "superiority" to the car, we have no reason for delay.

Having come closer to describing revenge, have we drawn any closer to understanding what it is about? Suppose Desdemona had actually been unfaithful to Othello. How does his killing her alter this? She still remains, even after her death, someone who betrayed him.

Scheler says,

*There will be no ressentiment if he who thirsts for revenge really acts and avenges himself, if he who is consumed by hatred harms his enemy, gives him "a piece of his mind", or even merely vents his spleen in the presence of others.* (Scheler 2015, p. 7)

The aberration that is ressentiment is a consequence of a revenge never taken. It is, says Scheler, an enduring poisoning of the mind, caused by a systematic repression of the revenge never taken, leading to spite, envy, and the impulse to detract (Scheler 2015, p. 4). This suggests that with an accomplished revenge something is put to rest, something completed, the tat has been satisfied with the tit. But this suggests that the original harm

has been somehow and by some measure healed. If we are to come closer to understanding revenge, we must draw closer to what is harmed in revenge and how revenge can heal it.

The puzzle of revenge is what exactly is put to rest and why. We will suggest in what follows that harm can be regarded, in Marion's sense, as a saturated experience, and that the ensuing violence and revenge are an attempt to address the overwhelming characteristics of a negatively saturated experience.

## 2. Saturated Phenomena

Marion's saturated phenomena are distinct from the constituted phenomena of the everyday world. What is important about these saturated phenomena is that the phenomenon given to the intuition run ahead of the intention. It exceeds the concept (Marion 2002a, p. 199). Because what is given "cannot be aimed at", it cannot be foreseen (Marion 2002a, p. 199). Inasmuch as a whole cannot be synthesized from a "summation of partial quantities" (Marion 2002a, p. 201), our gaze produces "amazement" (Marion 2002a, p. 201). Nonetheless, while something is visible, "our gaze cannot sustain" (Marion 2002a, p. 203) it. The quality of the phenomena is received as "excess", too much. "It fills without measure" (Marion 2002a, p. 204). "Intuition gives too intensely for the gaze to have enough heart to truly see what it cannot conceive, only barely receive, or sometimes even confront. This blindness stems from the intensity of intuition and not from its quantity" (Marion 2002a, p. 204). In its absoluteness, "it evades any analogy of experience" (Marion 2002a, p. 206). The saturated phenomenon refuses to let itself be regarded as an object precisely because it appears with multiple and indescribable excess that annuls all effort at constitution (Marion 2002a, p. 213).

Espen Dahl worries that Marion, in trying to argue for a pure given, utterly free of the familiar, would result in not being able to confer any sense whatsoever on the phenomena (Dahl 2010, p. 122). Dahl's concern is for the manifestation of the holy. He argues that "the mystery unfolds in the interval between the alien and the familiar" (Dahl 2010, p. 121), and that is the everyday.

The holy for Dahl is more than the alien. That something else is derived from the everyday, from a sedimented tradition. This must be the case not only for the holy, but also for all saturated phenomena as well. The Face, the notion of an historical event, or a painting all take up residence in the everyday familiar world, even as they serve as carriers of the saturated phenomena.

Dahl's description of how saturated phenomena are made "visible" remedies Marion's (perhaps) overemphasis on the invisibility of the saturated phenomenon. Dahl imagines instead a necessary marriage of the familiar and the alien, a kind of incarnation, where the "otherness is always incarnated in the same noema [of the familiar], thereby preventing a leap to another familiar noema or to the abstract other" (Dahl 2010, p. 293). The typical response in the everyday is to replace one familiar noema with another (Dahl 2010, p. 290; Marion 2002a, p. 211). This marriage prevents the leap. Dahl argues that "the whole noema will accordingly be unstable" (Dahl 2010, p. 293), so that "the entire noema starts to waver between the familiar and the alien" (Dahl 2010, p. 293).

Because of the importance Dahl places on the everyday familiar in the appresentation of the holy, and similar saturated phenomena, the role of analogy has to be reconsidered. Marion insists that the saturated phenomena are without analogy. Dahl insists that the holy, and we might say other saturated phenomena, are awakened through a passive synthesis of association (Dahl 2010, p. 294). As such, certain prior and sedimented traditions or understandings can serve as gateways, to use Schütz's word, into the saturated phenomena. This suggests that entrance into saturated phenomena is mediated by analogous phenomena and intentionality, that is, that there is a normativity and criteria of confirmation that constitutes an entrance.

Dahl sees the Holy as an "interruption" of the Everyday. As such, the Holy emerges from within the Everyday, but is nonetheless alien. He says, "This alienness is not tantamount to absence or infinity, but is rather positively given to experience" (Dahl 2010, p. 300).

We might say, then, that the saturated phenomena of the Holy hangs onto the coattails of some everyday intentional object. The horizon of the intentional object is exploded in the saturated Holy. Nonetheless, the horizon is required for the holy's disclosure; by its being exceeded, the holy is manifest through the visible and familiar.

It seems that Rudolf Otto would agree with Dahl. He says something similar about the numinous object, viz., that "the methods by which the numinous feeling is presented and evoked are indirect" (Otto 1936, p. 64). "[A]ny form of the numinous consciousness may be stirred by means of feeling analogous to it of a 'natural' kind, and then itself pass over into these, or, more properly, be replaced by them" (Otto 1936, p. 66). Otto argues that the numinous experience, which bears significant similarity with the experience of saturated phenomena, is "stirred" up by analogy with what can be encountered in the familiar, everyday world.[2] The numinous, however, as for saturated phenomena, are "perfectly sui generis and irreducible to any other" mental state (Otto 1936, p. 7). As such, all analogs must ultimately fail.

Otto argues,

*Whatever has loomed upon the world of his ordinary concerns as something terrifying and baffling to the intellect; whatever among natural occurrences or events in the human, animal, or vegetable kingdoms has set him astare in wonder and astonishment – such things have ever aroused in man, and become endued with, the 'daemonic dread' and 'numinous' feeling, so as to become 'portents', 'prodigies', and 'marvels'.* (Otto 1936, p. 66)

As such, the analogy between the familiar and the saturated phenomena rests upon the analogy between the finite and infinitizing of the finite, between the unsaturated and the saturated unsaturated, between a "shuddering" and something that is more than "natural fear" (Otto 1936, p. 15). It is this "more" that is immeasurable. The analogy works by taking what we can conceive, we can constitute and make sense of, and somehow adding the infinite ineffable to it; otherwise, we would not be able to make any sense of such expressions as the "wholly other".

The saturated are more than simply alien. The presentation must overwhelm you. It is not under your control. You are either attracted or seized. For saturation, there must be a sense of being held in thrall. As such, time may seem to slow down. This is where the sense of an infinity comes in. In saturation, we might say that there is an experience of an infinity, positive or negative, that holds you and prohibits your finitizing it and bringing it into focus. It overflows you, positively or negatively.

Hence, despite the inability to capture and constitute a saturated phenomenon as an intentional object, in order for it to be visible at all as anything, some aspect of a familiar horizon must be manifest.

### 3. Negative Saturation

Can we associate a valence or intensity with saturated events or phenomena?

Historical events and paintings are mentioned by Marion as saturated phenomena. The historian investigating the French Revolution would encounter the overwhelming depth and richness of such an event. Even so, their sensed distance from the saturation is great. They are free to abandon or engage with it as they wish. There is a significant element of mastery or control over their engagement. The same is true for a painting. The situation is different for the Flesh or the Face. The phenomena of the flesh are irrevocably our own. We cannot share them or escape them. There is no choice about them. As such, our encounter with the phenomena of the flesh is inescapably intimate. The same is true for the Face, which is why Marion refers to it as a burden (Marion 2002a, p. 232).

According to Brian Robinette, "Marion says very little about the saturating phenomena of loss and grief, suffering and evil, violence and exile". He suggests that "the experience of radical negativity might well be a kind of saturated phenomenon"(Robinette 2007, pp. 86–108).

Drawing on the work of Edward Schillebeeckx, Robinette suggests that "Evil, death and suffering—particularly massive innocent suffering—are refractory for thought. They flood and overwhelm the concept, disrupt our metaphysics, baffle our attempts to construct durable theodicies and universal theories of history".

Summarizing his understanding of Schillebeeckx, Robinette says,

> *Must we not also speak of negative bedazzlement, the fragmentation of horizons, the traumatizing of a language that cannot properly name the radical mystery of suffering and evil? Schillebeeckx discerns in negative contrast experiences unmistakable and distinctive traits. Whereas the aesthetic encounter is enrapturing, 'goal-less', and playfully expansive; suffering touches off a critical, cognitive force for its overcoming. The former invites contemplative lingering; the latter urgency for transformation. The former is rooted in manifestation; the latter is dialectical in its yearning for a time 'to come', freedom from what assails it.* (Robinette 2007, Op. cit)

Otto's notion of the possibilities of the numinous allows what might be regarded as both positive and negative elements. The wholly other is both daunting and fascinating. These two poles are generally always present, even in what he calls the daemonic–divine (Otto 1936, p. 31), where one is filled with horror and dread, tremble before it, and is utterly cowed and cast down, and yet one is still captivated and drawn toward (Otto 1936, p. 31). However, in the case of "daemonic–*dread*" only one pole remains: that of terror and feared wrath. One is more bound in this case than drawn or fascinated. In this case, one is face to face with the daemonic–dread numinous and by its very overpowering, inescapable binding.

Dahl, when speaking of the holy, said that the saturated phenomena oscillated between the familiar and the alien. It seems that there would be, in this regard, a difference between positive and negative saturation. In positive saturation, there is time and place to linger. There is sufficient phenomenological distance for the apperception to withdraw to the familiar or fall into the saturation. But with negative saturation, the Situation is encompassing and suffocating. There is not a sense of Levinas' freedom, but rather of enslavement. In both positive and negative saturation, one is enthralled. In the former, one might be more allured, while in the latter overcome by dread. This would suggest that in positive saturation, one experiences a kind of waiting or a resting, while for negative saturation, one desires to flee. It would seem that the temporal experience of a waiting or resting is different from that of fleeing. In the former, time might be experienced as slowed down, almost timeless, as if we were not aware of time passing, whereas in the latter time would be experienced as stopped or held up, as such a heightened sense of time, even a frustrated urgency.

It seems then that none of the saturated, numinous, or alien are in themselves positive or negative. They can take on aspects of negativity, positivity, and even perhaps a neutrality. What determines the valence of the excess phenomenon is something more, something added by context and what is familiar and Everyday.

We are ambiguous about possibility. It takes on the flavor of Kierkegaard's dread and Otto's numinous. It can be both alluring and terrifying. The alluring part is positive, the terrifying negative. Perhaps it is not the saturation itself that determines whether an event is positive or negative, rather it is the understandable context in which the saturated event is given. If the saturated are indeed non-objectifying, then it seems that it is difficult to make much of them. We experience them as more than we can comprehend or grasp. As such, we experience their beyond, their unknowingness, their openness, their "infinity". Following Dahl, we might say that all saturated phenomena are given in the context of something familiar, and, as such, understandable and objectified. The alien or saturated aspect of the experience bursts the bounds of the familiar, presenting to us as something alien.

Negative saturation is associated with death and dying, with grave illness, for example, positive saturation with encounters with the other, as in Levinas' face. But the context here,

the familiar setting, is different in each, and it is that which determines the valence of the saturation.

## 4. Harm

Since harm is fundamental to any understanding of revenge, we first examine harm. We do not seek revenge on another unless we apperceive that we have been harmed in some way. We constitute harm or being harmed out of some perceived change to which we are intimately associated. But it is not merely any change that is required. Harm entails that we judge the change as lesser in some sense. The changed state is, in accordance with come criteria, diminished. There are degrees of harm as well. As such, harm requires a complex of interrelated meanings and a presumed axiological structure.

All of this suggests that harm is the loss of something, something missing. It is a negation, a making of something nothing. We can imagine that this is the result of constitutional disappointment. It is not merely that the constitution is unfilled, rather it is that it is disappointed, which is to say that it has been confirmed that something is amiss, something is missing.

The distinction between unfulfillment and disappointment is illustrated by going to meet Pierre at the airport. As long as you do not see Pierre, as long as you are at the airport waiting for him to disembark from the airplane, the constitution of Pierre is unfulfilled. We have constitutionally confirmed that we are at the airport and that the plane is to arrive at such and such gate. We have fulfilled all the criteria necessary for the arrival of Pierre. Still, the arrival of Pierre is unfulfilled. The constitution of Pierre's arrival is not disappointed until the very last passenger has disembarked, and none of them were Pierre.

In harm there is a disappointment. But there is never a disappointment without anticipation. We anticipate the arrival of Pierre, which is to project into what is not-yet, to complete the constitution of an intentional object that is not-yet. In the anticipation, we fill in what is not-yet confirmed. It is only by comparing the anticipated intentional object, which is guided by the noema of an unfulfilled one, that an up-dated constitution can disappoint.

As such, harm requires some intentional object with its associated noema, an anticipation of an unfulfilled constitution, and a disappointed constitution. The disappointed constitution is of some intentional object that is "less" than the anticipated object. It is lacking something of the anticipated intentional object.

As such, a cut finger is judged to be harmed because a finger that is cut is judged to be "less" than an uncut finger. A "cut finger" in and of itself is not harmed. It is simply another intentional object, and not yet a "harmed" finger. In addition, there must be the disappointment associated with the "cut finger" in order to constitute the "harmed" finger.

All apperception of change requires the anticipation and the disappointment of that anticipated intentional object. We want to go fishing, but dark clouds fill the sky and rain begins to fall. Suddenly, the clouds break up and the sun begins to brightly shine. We apperceive a change and disappointment, but we do not apperceive it as harm. Instead, it is perceived as just the opposite, good fortune.

For this reason, one additional feature is required for "harm". A "cut finger" is not harmed, even if it is a disappointed "uncut finger", unless we supplement the disappointment with a judgment, and evaluation, that this is a "bad" thing, and in some sense undesired.

Having come this far, we cannot be done. It still remains to have some understanding of the "desired" and "undesired", the "good", the "bad", and the "ugly". There is associated with these "negatively" apperceived events a wish, even a strong wish, that it was not so. And this entails that we can imagine it to be different than what is given to us, a counterfactual capability. The very fact that we are wishing entails that we have the sense that the state of affairs is not up to us. Instead, they are thrust upon us, even if we believe that there is something we can do to change it in the future to the wished-for state. In this

sense, the "undesired" is like an illness that takes over our body, alienating us from our own body and what we regard as our own.

Māra Grīnfelde speaks of illness as a saturated phenomenon. If illness is a saturated phenomenon, can we say that all harm has the characteristics of a saturated phenomenon? Consider Othello. He was harmed, let us say by his shame, the shame of being betrayable, of having been betrayed. He was alienated from himself and regarded this new self as reprehensible and repulsive. And yet he cannot get past this new self. It is stuck to him, and yet it is alien, which lends itself to a kind of negative saturation, something that he cannot get his hands around.

Māra Grīnfelde argues that "the experience of illness in its essence is very similar to the experience of other saturated phenomena (Grīnfelde 2019). She says,

> *According to Marion's interpretation, intentionality always restricts the given intuitive excess (what we see, smell, hear etc.) into definition, essence or the sense of the object* (Marion 2002b, p. 108). *Intentionality thus aims **not at what one actually sees** (for example, the different play of colors) but at the **significance of the seen** (the meaning of a street sign, for example), which is important for practical knowledge (whether it is safe to cross the road here or not). Interpreted in such a way, intentionality, according to Marion, expresses mastery and control of the constituting subject [emphasis added].* (Grīnfelde 2019)

Intentionality aims to massage and manipulate what is given into something "significant" and "understandable". If that is the case, how are we to regard intentional objects, such as illness and suffering, that we cannot massage into a desirable form?

Understood as a saturated phenomenon, illness manifests itself as an "excess of intuition over intention/concept", which means that it manifests itself as an immediate succession of affections that overwhelm the subject's intentional activity.

Referring to a "cluster of experiences, such as pain, suffering, fatigue, nausea, anxiety, irritability, helplessness, sadness, itchiness", she says, that "According to Marion, these feelings are immediate or nonintentional, meaning that they are given in experience without any representative or intentional gap and refer to themselves alone".

> *The more overwhelming the affections are, the more disruptive the illness is—the less overwhelming the affections are, the less disruptive the illness is. This explains why some minor and transitory ailments such as the common cold do not necessarily disrupt our habitual engagement in a world—the affective givenness is not strong enough to overwhelm the intentional activity of the subject. The stronger the givenness of affections, the less functional the intentional activity and the less we are able to comprehend the world and engage in it in a habitual way.*

> *As the saturated phenomenon, illness happens to us, shattering the sense of the self as an autonomous subject. Faced with the excess of affections and the accompanying disruption of possibilities of perception and action, we experience ourselves as out of control, not mastering the givenness but witnessing its unfolding.* (Grīnfelde 2019)

Can we say, then, that, as for illness, harm is a saturated phenomenon? There are degrees of harm. We might say that a paper cut is a lesser harm. Inasmuch as we contain the harm, inasmuch as we can limit its scope and define its boundaries, do we rescue it from saturation? A paper cut is different from a slight or an insult. Saturation carries with it an aspect of being able to bind and limit the phenomena. Do we take a paper cut to be a "small harm"? We know what to expect and know that in a day or two we will have forgotten all about it. This is not the case for an insult. Its scope appears unlimited or amorphous.

Illness of the body (the flesh) is always experienced as me that is in pain. This is likewise the case for harm done to what we might call the self. And yet they are qualitatively different. A broken arm is different from leukemia. In both cases, we feel a kind of impotence. We cannot escape them. The former is visible. We can see the harm, and know something of how to lessen the pain and heal it. Leukemia is invisible; even its

symptoms are mysterious and hidden. We can give it a name so we can refer to it. What we are told is happening is that our bodies are in a sense betraying us, working against us, in this autoimmune disease. As such, it is slippery, showing as some kind of confusion as to what *our* body is, and what it is to be mine. I want to suggest that the same is the case for the wounding of the self.

When the "self" is harmed, it is not exactly clear what is harmed or how it may be healed. It is difficult to get at it and know how to set it aright. We can give it a name, such as remorse, shame, or guilt, and we can even try to describe what it is like to be in one of these states, but it can never be enough. We can never capture what it is for them to have a grip on us. There is something of the alien about them. They are ours, and that we cannot deny, but at the same time they are unfamiliar and foreign, as if we were possessed, and they have us. These harm the "self"; we only know that we have a sense of urgency about them, that we ought to relieve ourselves of them somehow. As such, they overwhelm us and, as such, come against us as negative saturated experiences.

We have perhaps been less careful than we need to be. Harm is different from suffering. That I am harmed does not entail that I am suffering. It is not so much harm that initiates revenge, but the apperceived suffering. Apperceived harm is necessary, but without the concomitant suffering, there would be no "necessity" of revenge. The harm of an illness results in suffering by way of my involvement with the harm. The harm, we want to say, by way of a negatively saturated experience, becomes suffering.

## 5. Violence

Revenge, at least the revenge we have in mind, is often violent: murder or worse is in the air. What is the origin of violence? Marion says that "[René] Girard instead identifies a recurring violence at each moment of the establishment or the reestablishment of the social community" (Marion 2020). René Girard understands violence in society to result from memetic rivalry. Marion suggests that this violence derives from the saturated presentation of the Other. Marion says of Girard:

> Girard takes violence as an acquired, original, and already given fact. Mimesis does not explain the emergence of violence, but defines and describes its allegedly primitive force: namely that of rivalry.
>
> Why does violence almost inevitably arise, when two egos confront each other face-to-face? In order to get right to what is most essential, let me summarize Lévinas' analysis: "The face is present in its refusal to be contained". By refusing to be contained and to be constituted as an object, the face wards off any taking possession by my intentionality and thus imposes itself as a phenomenon that is expressed from itself". (Marion 2020)

Marion replaces Levinas' Hegelian lord–slave self-consciousness with "a mimetic *rivalry* (in Girard's sense) because the two egos purport to keep constituting each other at the same time and reciprocally", which results in constituting the face as an object and a consequent rivalry.

Does this Hegelian image of a matured selfhood entail that one self would not do violence to another? There is a kind of mutual dependence, each constituting the other. Surely much violence is committed as a result of us treating others as things: things that can be easily envisioned and configured as enemies and a threat to our selfhood. But, just as common is violence between those that are dear to each other, between what we might call "loved ones". Here, two or more are closely knit together. The violence occurs because the other is almost indistinguishable from our selves. As such, we feel compelled to deny them their freedom and independence. Is this, then, to no longer treat them as selves?

Violence accrues, says Marion, from my habitual mastery of my world to make a thing of it. Since a genuine encounter with the face is rare, at best it seems we can say that our ability to objectify is necessary for violence, but not sufficient, unless we are to define violence as mere "objectification". Rather it seems that the "objectification", the abstraction of the other, allows the other to participate as a theme in some drama where

violence plays its part. The "objectification" prepares and lays out a clearing for possible subsequent violence.

## 6. Revenge and Pairing

Pairing, according to Husserl, occurs passively with regard to things that are similar. He discusses this at length in *Cartesian Meditations* (Section 51) in attempting to uncover the constitution of the other subject through the similar bodies. Here, in revenge I suggest there is a pairing between the "loss" in harm and the guilty. The question is whether that passive pairing necessarily results in revenge. And if that is the case, what then determines the kind of pairing? Can there be a pairing towards forgiveness, something that is beyond the scope of this essay? I will suggest later that it has to do with the form of life, perhaps held in customs and myths that provide the shape of our world.

The pairing between the loss and the guilty, is that the right pairing? They are similar in and through the loss. That is why there is a guilty one. She is linked with the loss through the guilt. Pairing says Husserl, in Section 51 of *Cartesian Meditations*, is a "universal phenomenon of the transcendental sphere (and of the parallel sphere of intentional psychology)". "Pairing is a primal form of that passive synthesis which we designate as 'association,' in contrast to passive synthesis of 'identification'" (Husserl 1999).

> In pairing association the characteristic feature is that, in the most primitive case, two data are given intuitionally, and with prominence, in the unity of a consciousness and that, on this basis—essentially, already in pure passivity (regardless therefore of whether they are noticed or unnoticed)—as data appearing with mutual distinctiveness, they found phenomenologically a unity of similarity and thus are always constituted precisely as a pair. If there are more than two such data, then a phenomenally unitary group, a plurality, becomes constituted. On more precise analysis we find essentially present here an intentional overreaching, coming about genetically (and by essential necessity) as soon as the data undergo pairing have become prominent and simultaneously intended we find, more particularly, a living mutual awakening and an overlaying of each with the objective sense of the other. This overlaying can bring a total or a partial coincidence, which in any particular instance has its degree, the limiting case being that of complete 'likeness.' As a result of this overlaying, there takes place in the paired data a mutual transfer of sense—that is to say: an apperception of each according to the sense of the other, so far as moments of sense actualized in what is experienced do not annul this transfer, with the consciousness of 'different'. (Husserl 1999)

The pairing comes about, in the case of the other subject, "when the Other enters my field of perception". As such, it comes about upon perception of the body of another. So, it begins at least as a bodily relation. It does not produce an identity but rather an *Einheit als Paar*, a unity as a pair. They remain distinct as a dual pair.

In what ways they pair matter, and there are degrees of pairing. Attributes of the two are not necessarily those that are essential to each, as this is too close to one or the other. In the case of revenge, the loss and the guilty are both associated with the loss, and yet the loss is not intrinsic to either one.

Frederica Giardini remarks:

> In Ideen II (§ 56b) there is another way to conceive the Pairing in the light of passivity: in the past a connection occurred and in the present there is a tension to repeat that connection, but not exactly the same, an analogon of it. It is a law of the motivation. Given in the past, Pairing structures the present too. (Giardini 2003)

We intend to show that it is by pairing that the one harmed, the harm, and the one guilty are associated. This pairing makes it possible for conflicting, even contradictory, intentional objects to be brought near, thereby constituting the painful, irreconcilable loss that motivates and directs revenge.

## 7. Expectations and Anticipation

We are forever trying to fix the future. We rely upon a certain kind of world, even a certain kind of person, in order to do this.

Expectations, anticipation, hope, and the like all are temporal notions. How do we come by them? Trees, we have well learned, here today are likewise there tomorrow. We have an expectation about trees, that they possess this enduring property, even that the world is such that it is commonly satisfied. So we can anticipate hanging a swing from a tree tomorrow, and it still be hanging there years from now. The same is true of our daughters and eyes. We expect our daughter, here today, to be here tomorrow, so also with our eyes. But this is not always the case. Our daughter dies. We lose our eyesight. How did that happen?

What if we did not have such expectations? Suppose our daughter is more like a soap bubble, our eyes like smoke? We would not be surprised if we woke up to find our daughter gone, now dead, or our eyes no longer functioning. We do not attach ourselves to things such as soap bubbles and smoke the way that we do to daughters and eyes.

We have a different relationship with the enduring than the eminently transient. With the enduring we can invest ourselves. This very investing indicates the vulnerability of these enduring. The enduring lie somewhere between the transient and the eternal or immortal. What shall we call them, these that are in between, for this in between is important for our enduring. The enduring are vulnerable.

It is by and through our expectations and disappointments that our finitude is worn sore. Were it not for such as these, we might be unknowing, perhaps even content. It is the temptation of the perishing–durables that undoes us. As enduring, they can be expected, anticipated, hoped for, and stockpiled. As perishing, they are vulnerable, must be protected and preserved, and ultimately replaced. It is a world that is forever decaying and renewing. It is in this sense a temporal world, as we ourselves are.

There are those transient that we honor, exalt, and regard highly: a rainbow, a delicate flower. There is no owning of them. No one owns the transient, even while she might praise them. Our relationship with the perishing–enduring is different from that with the transient. I suppose someone could hasten the demise of the transient. But it is with the perishing–durables that we become attached.

What is this attaching? Attaching is like a pairing. We can become attached to an old shoe, a shrub that we pass by regularly. It becomes integrated into our "world", and as part of "our" world, it is family. It is familiar. We become paired with it. Harm to it is a kind of harm to us, almost a threat to our *heimlichkeit*. It is more than valuing. We may not value the shrub. We may not water it or care for it. Still, we are paired together, and its loss is a loss for us. The familiar is our defense against the "ravages of time", of its decay. It is this enduring aspect that stands in for the enduring, while still aware of the pain of perishing. Our attachment, our pairing with the enduring, is a manifestation of our enduring. Inasmuch as they endure, so too do we, by our pairing, endure.

We note also that in the constitution of the enduring, we have to appropriately select what attributes are to identify the entity. These are the noema of the enduring. We say they are essential or the essence of the thing, but really what we are doing is selecting attributes that do and can endure, and this is because we are after enduring objects.

This is important for pairing. In the pairing of two, it is not an identity. Rather it is a relationship of similarity (Husserl 1999). The attributes united in the pairing must be likewise enduring. That suggests that we do not necessarily unite essential features of each. The essential are perhaps too close to each one (Giardini 2003).

Can we begin to see, then, what revenge is about? Is it no more than a common manifestation of our relationship with the enduring–perishing? In the perishing, the losing of the enduring, we suffer our own existential fragility. Revenge is not conceivable without the perishing–enduring and our attachment to them. It is only by this pairing–attachment that suffering can arise.

## 8. Disappointment

Expectations of the not-yet rely upon enduring objects and noema. These are mental objects. We could call them ideas. I am not certain how rigidly rational we should be about such objects. We all know what a hammer is and what it is used for. But few could provide a definition of one. These mental objects are probably associated with notions of a paradigm and our ability to recognize similarities. It does not require a dictionary, but rather embodied experience.

With these expectations, we have also notions of disappointments. The expectations, like intentional objects, have their criteria of satisfaction, confirmation, and disconfirmation. A terracotta version of a hammer may look from a distance very much like a hammer. But we soon realize, if we try to use it as a hammer, that it is not a hammer. It failed our noetic accomplishment of a hammer. As such, we are disappointed.

In the disappointment, the unfulfilled constitution, something else is being constituted, for something is always being constituted, even if ambiguous or ambivalent, a contest of intentional objects. Here is constituted, at least, disappointment.

We need to distinguish two kinds of disappointment; one matters, the other does not. Suppose I am walking along a street, and I think I see a man standing outside a store, but I cannot be sure. The person is decked in some kind of fantastic attire. As I approach and come closer, my constitution of a man is disappointed. It is not a man, but a mannequin. It does not really matter to me whether it is a man or a mannequin, but, nonetheless, my original surmise was incorrect, thus my disappointment.

On the other hand, if my daughter had been out on New Year's Eve helping to transport those who have had too much to drink safely home, I expect her to arrive safely home some time after midnight. I anticipate her arrival. When I see her car approaching the house, I have the beginnings of a confirmation of an intentional object: her walking soon into the house. This constitution is disappointed by seeing instead a friend exit the car and then being told by her that your daughter has been rushed to the hospital in critical condition. This is a disappointed intentional object (her arrival home) that matters deeply. It is unfilled and replaced by one that you did not expect, one that is now confirmed as you stand over her in the hospital room, her life now impossibly gone.

We do not have absolute noetic control over the intentional objects that are constituted. It requires certain embodied practices and capabilities, but the associated noema of the intentional object constrains what can be constituted by certain criteria of confirmation and disconfirmation. If this were not the case, we could constitute anything from any phenomenon, even a peanut butter sandwich from the Grand Canyon.

In some sense, every constituted intentional object matters. We would, after all, not have a meaningful intentional object if it did not matter. We would have no name for it. We would not notice it at all. So there must be degrees of mattering. Why is it that a daughter matters more than any blade of grass?

The carrier of what matters and its intensity must have something to do with a kind of memory. These memories are not immediately available phenomena in the stream of consciousness. Of course, this is true for any intentional object. The constitution of any intentional object cannot be accounted for in immediate experience alone. It requires a pre-existing noema and noetic practice to realize the intentional object.

Zahavi says of Husserl,

> *Every intentional experience is an experience of a specific type, be it an experience of hoping, desiring, remembering, affirming, doubting, fearing, and the like. Husserl called this aspect of experience the intentional quality of the experience. Every intentional experience is also directed at something, is also about something, be it the experience of a deer, a car, or a mathematical state of affairs. Husserl called this component that specifies what the experience is about the intentional matter of the experience.* (Zahavi 2003)

How do such qualities or attitudes become attached to an intentional object? The intentional object of "my daughter" has a different feel than "a daughter". One does not seek revenge for "a daughter", but instead for "my daughter".

The daughter who leaves the room or goes to visit her aunt is not present. We have this non-present daughter in a way different from when she is present. The noema–noetic structure for "my daughter" remains intact. As such, I can miss her since I cannot accomplish the noetic procedures for her to be "present". Nonetheless, still remembering the noetic practice (stroking her hair, gazing upon her playing), I can see them as empty practices that constitute her as "missing". I may have these in memory, knowing that it is a memory and qualitatively different from a perception. The act of stroking her hair is unfilled, as such stroking brings her into presence.

Is it possible that such unfulfilled noetic acts are present in the act of revenge? This suggests that the act of revenge is not only (if at all) meant to punish, or not so much to punish the guilty, but to restore the daughter. The empty noetic acts of, say, stroking her hair, brings the daughter into presencing in both perception and in missing her. By recalling the noetic confirmation of stroking her hair, even though unfulfilled, it nonetheless constitutes the missing child. Without it, there would be no missing child.

Consider now confronting the drunk who killed her.[3] The disclosing of his present body and the prior belief that he is responsible for their daughter's death activates the death, the irrevocable "missingness", of their daughter. Let us say that they never met this person. As such, he is a stranger, until they told that this is the man who killed their daughter. Once that connection or pairing is established, it is sedimented somehow. Passively paired are his body, his guilt, and their daughter's death.

They are not paired in a way that might be similar to their daughter and a stuffed animal that she had grown up with. They might want to handle the stuffed animal, even stroke it, and in this way bring their daughter near in a way that it not unlike stroking their living daughter. This is also, it seems, another instance of constituting her being "missing": the recognized unfulfilled constitution.

The pairing of the guilty one and her daughter is perhaps more like the pairing of an ex-wife's body and a bitter divorce, or of a particular car and a horrible accident. Pairings result in intentional objects. As such, they have a certain quality and character. It seems that they can be positive, negative, or more neutral. We might say they could be pleasant, unpleasant, endearing, frightening, indifferent, or incidental. It seems that such possibilities are possible because of other associations, other meanings. Why expect that the pairing of the guilty one's body, his guilt, and their daughter's death would, upon perceiving the body of the guilty one (for his guilt and daughter's death are invisible), engender anger, etc.? Why attach these three pairings with rage, punishment, anger, revenge, harm, and suffering? It is, nonetheless, clear that harm is connected with the pairing.

We have already established that the death of the daughter is paired with harm, grief, and the missing daughter. The guilty man has harmed both daughter and the parents. Here, at least, we have a connection with harm. Harm is in the air. If we could see why there is a tit for tat, we might see how revenge sees that the tat of harm will result in a tit on the guilty one. Remember, the principal harm is the missing daughter. We can noetically accomplish the missing daughter by pairings with things already paired with her. Surely the sight of the killer accomplishes also the missing daughter, a pain and suffering that is now not only about the missing daughter, but also about guilt. Guilt assigns blame. In particular, it pairs with the missingness of the daughter, not merely that the daughter is missing. It is that which makes her missing. And if it is that which makes her missing, might not it also make her not missing? The guilt of the killer pairs the killer with the missingness of the daughter, as would the blinding of blindness.

So, it is possible to pair the embodied killer with the missingness of the daughter, and then to pair the missingness (the ability to make missing) with the returning of the daughter. There are two questions in this regard. First, this may seem farfetched. Of course, we must never forget that the revenge is intensely related to the missing daughter. The missing is prominent. If this is the case, harming the killer will not obviously return the daughter. Indeed, it appears detached from the daughter entirely, just as in the case of

blindness. The killer might seem to be a kind of magician: making the daughter disappear. As a magician, perhaps he can return the daughter.

Consider the revenge of frustration. Imagine that a nail will not go into a board, stalling progress on some project. In frustration, we smash the board, damaging part of the project. The board, the nail, become this monster that is holding up progress, halting time. We have to defeat it. Kill it so that time can go on, like the witch in C.S. Lewis' *Narnia* who keeps it always winter. The nail and board, like the killer, become associated with the cause and what brings about the situation. The situation here is the stopping of time and the progress of the project. For the daughter, it is the killer who has halted the daughter project. If he is beheaded, the project can go on, we might "reason".

In what sense can either the board or the daughter project go on? We have destroyed the board project, and the daughter is still dead. Are they not? Do we really believe that the board project can now go on in the same way that it was before our act of revenge? Do we do the same with the daughter project? There still remains this puzzle: in the pairing of the killer and the missingness of the daughter, why try to harm the killer? Are there no other possibilities?

What is ultimately so frustrating and counterproductive about both is that neither act of revenge actually succeeds. The project has to be started all over again, and the daughter remains dead. In a sense, we were sadly mistaken.

There are two other possibilities. The first is that our act of revenge is aimed at addressing something else besides the missing daughter. The second is that there are other ways of responding to this pairing of missingness of the daughter and the killer, including forgiveness.

Another possible approach to the daughter and the board is to imagine that it has something to do with re-establishing our mastery. In this case, the killer has mastered us, just as the nail and board have. The project, the daughter, are less important than our mastery. So by destroying the board, we express, we reaffirm our will, our mastery over something. Okay, if we cannot build a project, we can destroy the board, even the project which, in a sense, has mastered us, so too with the killer. He has mastered us in killing the daughter. I am weakened and overcome. For the sake of my sense of mastery, I master him. I can master something. This picture returns to Schütz's I-can and I-can't in the working world as the interpretive template. Such a perspective appears to be in line with Marion's notion of the origins of violence. In the I-can and the I-can't perspective, it is the I-can that is at stake and central. The guilt of the killer may still pair the killer with the daughter's loss, and with the I-can't. We could crudely say that my pride has been injured, with an emphasis upon "my".

So the two pictures appear starkly different. The first attends to the loss of the daughter, while the second attends to me. According to this understanding, they both want to restore. In the first it is the daughter, while in the second it is my "pride", my sense of mastery.

Can we say, then, that there is a difference in focus? Let us just say that pride of some sort might motivate the avenger in the I-can and the I-can't, in the pursuit of mastery.

## 9. Approach to Saturation

"Your daughter is dead", you are told, perhaps by the police, people you do not even know. The intentional content of what is said is plain enough to you. We are dealing here with intersubjective elements: "daughter", "dead", even "your" might have an intersubjective element, expressing aspects of experience that are shareable. What is not shareable, however, is the intentional quality of this "your". The same is the case for this "death", which has both shareable and unshareable aspects. Nonetheless, this "your daughter is dead" is an intentional object. It is in the intersection of intentional horizons by which the full synthesis is drawn by way of similarity and association. This "your" is found in the intersection of horizonal meanings linking the intersubjective reality with understandings and experiences of "self", whereas the intersubjective, or perhaps "objective" reality is "nearly complete"; the more "subjective", unshareable reality appears

more unbounded, its horizon more open-ended. This distinction is drawn in sensed possibility. The possibilities of a "tree" appear, for the most part, to be quite limited and "well-defined". This is the not the case for "your daughter", and this is not simply because she is "human", for we would find the same for your next-door neighbor.

For the most part, much of our experienced life in the natural attitude appears, we might say, "static". Objects, even experiences, come already made and prepared before we and they arrive. When we encounter the vast repertoire of intentional objects inhabiting our meaning landscape, horizons of possibilities are activated. The "full" horizon of possibilities is not activated, but only a small subset. A "tree" we encounter on a family drive in our car is vastly different from the "tree" that we try desperately to avoid when we lose control of our car. In the latter case, the perceived scope of the tree's horizon is vastly expanded.

What we begin so see, here, is the central importance of expectation and anticipation in apperceived experience. The "tree" we see while traveling along a country road is associated with a very different expectation than the one we are desperately trying to avoid.

Suppose we encounter an acquaintance by accident. Immediately, we apperceive the situation. We anticipate a short conversation, perhaps asking what they are doing in town, etc. Suddenly, however, the conversation shifts in demeanor. He begins speaking about his family. We sense that the situation has changed because our expectations have been disappointed. As such, our expectations likewise change. We shift from one patterned gestalt to another. In both cases, however, we understand what is going on and how to get about in it. Both of these intentional forms have already been prepared for us before we were there together, in just the same way that "trees" and "country roads" were.

Suppose now, however, that there is an additional shift in the form of the conversation. He breaks the bounds of the previous intentional form and begins to tear up, quickly leading to weeping uncontrollably. Now we begin to feel uncomfortable. We are not sure what to do. Perhaps we do not even know this person very well. We feel uneasy. Essentially, we lack an intentional form of engagement to constitute an understandable situation, and for this reason we feel lost, perhaps even overwhelmed. We might even begin to seek an excuse to flee the situation. This is the approach to saturation.

In each engagement with our intentional world, there is a calling. It calls to us to engage with and in it. We respond to that calling in intentional and meaningful ways; and that means having some intentional form that can be accomplished by our noetic activity. Where we are unable to constitute or "objectify" (to use Marion's term) this experience, it confounds us, producing a kind of uneasiness; we might say an *unheimlichkeit*, something that can be seen in experiences as commonplace as embarrassment.

So it is for the "death of my daughter", which explodes the horizons of unsaid possibilities. As such, it likely strikes us as saturated. No longer can we adopt a natural attitude regarding "objects" as "over there". Instead, the calling has us and engages us, in Marion's sense constituting us. The distance between us and "experience" collapses; we are unable to flee or back off. It is the sensed expanding of the horizon that discloses the calling; and this is sensed as new situations are disclosed to us and their related expectations, through previously established understandings and their associated noema.

It is in the recognition of disappointment of expectations that horizons are expanded and the calling changes. The "your dead daughter" breaks the bounds of all expectations; and it is this "shock" that induces the saturated experience. Unlike the accidental encounter with an acquaintance, one cannot flee the situation, as we cannot for serious illness. By the pairing of association and similarity with understandings of your "self", even your embodied "self", you are unable to flee, no matter your desire to do so.

Saturation, according to this understanding, requires the familiar. It requires that there be expectations and anticipations, with their associated disappointments. For without these, there can be no "shock" of surprise, no sense of being at 70,000 fathoms, unable to feel the bottom (Kierkegaard 1992). Where there is no "shock" of surprise, of the utter failure of the constitution of expectation, there is no "constitution" of the saturated experience. It

seems, then, that all saturation is engendered by disappointed expectations, followed by a deepening engagement or calling into a place of saturation wherein we cannot find our footing, which means that we have no extant understanding of the situation that forms and guides our getting about in it. It is, plainly, alien and unfamiliar.

We might imagine, in pursuing this Heideggerian notion of understanding[4], that there is a possible situation where we have no, or very limited, understanding. In this case, we have little or no idea how to proceed. We have, we might say, no expectations. We simply lack sufficient understanding of the situation. This having no understanding, and no consequent expectations, is, nonetheless, experienced as something. It seems that in both this case and the case, previously discussed, wherein the horizons of our expectations have been exceeded, that there is a sense of the infinite. In the latter case of the surprised saturation, there is an experience of "excess", of too much. This is possible only because there is an experience of something being exceeded. In the former case, where understanding and expectation are lacking, we are dumb and silent. It does not seem that there is an experience of "excess", even if it appears boundless and infinite. In any case, our attention here will deal exclusively with the exceeding of the familiar, and not the expectationless.

## 10. Vengeance

Vengeance, like forgiveness, is closely associated with what is in the past. Some harm has been inflicted on you, a harm that you blame someone for. You are interested, however, in more than mere blame. You want to "get even". This vengeance is similar to mere blame in that the harm that you inflict upon another in "getting even" will not actually remedy what has happened in the past. If someone ran over your daughter while driving drunk, any vengeance you obtain will not bring your daughter back to life. It is not like restorative justice. Generally, if someone steals your car and it is returned in the same condition as when it was stolen, we are satisfied. It is possible, however, that we feel something more is required. We feel more than inconvenienced. We feel violated. And this violation cannot, like the car, be restored so easily.

This is where vengeance comes in: where something is harmed that cannot be readily restored. Vengeance lives close to saturation. There is an excess, something we cannot get our hands around that is part of the phenomenon. It has the sense of an infinity, something unbounded. This is in part because what the revenge seeks is something intangible. It is indefinite.

We can see this in the case of a woman that was raped. She has been harmed in intangible ways. It is not something she can point to or weigh. She cannot carry it around in a bag or even purchase it. It is both positive and negative, negative in the sense that something is lost and experienced as missing. That might be her sense of safety or well-being. But it is also positive since it is something that weighs on her. We might say that it is the phenomenon of her life that has become negatively saturated. This saturation is reflected in a *Befindlichkeit*, a mood, that colors all her involvements with the world, even herself.

How is it that vengeance arises? We want to say that it is a response to the saturation. Vengeance is somehow intended to bring the phenomenon of her life back into focus. She now finds it difficult to constitute her home as a safe place, something she never even considered before. It was taken for granted, never doubted. She notices this difference, this loss, of what was. In her present constitution of her home, what she previously constituted is part of the current constitution. It wobbles between the old, familiar constitution and as a place where a rape occurred, and will not go away. As such, it will not settle down to a stable apperception. The chairs, tables, and rugs are all familiar, all components of what constituted criteria of confirmation that this was her home, a home safe and comfortable. But this alien element, associated with the rape, keeps intervening, and disappointing her constitution of that old familiar home, as if the house were haunted with alien creatures. As such, her home is constituted as a mixture of the familiar and the alien.

This alien something is not merely something like a stranger, introduced to you by an acquaintance, who happens to need a place to stay for a few days. Such alienness might be awkward or inconvenient. But such a stranger remains within the horizon of what a casual acquaintance would be. The rapist burst the horizon of any notion of what an acceptable human interaction would look like. Indeed, there is no definite horizonal noema associated with the rapist. He is literally lawless, meaning that there are no bounds to his conduct, no definite expectations or anticipations associated with the rapist as an intentional object. He is wild, unpredictable, formless. This is the very source of the saturation: the infinite openness that takes her by storm. Here, we recognize the saturation that arises through disappointed expectations, and the shock of their horizons being exceeded.

Can vengeance be of benefit here? Can vengeance make it possible for her to be able to constitute her home as safe and familiar? It is a kind of exorcism: she must be able to get him out of her head, for surely he is always there, even if she knew he were locked away. The house is haunted. Were that the case, she might move, but would that help? He or someone like him always remains lurking, haunting her as if she is possessed. The exorcist calls upon the aid of a "higher power" to drive out what possesses the person. Vengeance, perhaps, provides her a means to manifest power over the rapist, as an attempt to restore her mastery over her life and the very constitution of her home. In this sense, it is felt as anger that heats up and invigorates our bodies and manifests a sense of power and strength. This anger is a new spirit, a new possession, that inhabits her body and takes on the inhabited demonic possession, perhaps rising to the point of driving it out. Inspirited with anger, she enters her home now with a renewed vigor, not as the frightened victim of his assault, but as a kind of warrior, ready to take him on. If this makes sense, the aim of the vengeance is to drive out and overpower the demon that resides within, the very alien demon that perpetually interrupts the constitution of her home as a home, safe and comfortable, as *her* home. Vengeance is a matter of constitution, to overcome the saturation and return it to the familiar.

As such, vengeance, like Girard's scapegoat (Girard 1986, chp. 3), has a ritualistic nature to it. The practice takes place in the everyday world with familiar objects: the rapist, the police, the jail, and who knows what else. The point of the practice is to humiliate and finitize the rapist. The outward practice's aim is, however, spiritual. Were it not so, the outward practice would lose its meaning and value.

It still remains to consider how it is that the ritual cleanses the soul. By this we mean to ask how it is that this embodied ritual serves to enable a reconstituted home, one that is now (again) familiar and safe. The saturation introduced by the rapist is not under the mastery of the woman. As with all saturation, it overwhelms her ability to master the constituted object. The rapist remains other, radically other, an "infinite" other without bounds, an instance of negative saturation.

If the problem is the infinite, that is, the excess, it would seem possible that the rapist could be reined in by familiarizing herself with the rapist. That is, if she could come to know the rapist, in a sense to humanize him, his infinite stature could be finitized, as such, his powers limited, his reach and desire constrained by familiar horizons. Lacking this rare possibility, the physical overcoming of the rapist, one could imagine that the humiliation and humbling of the rapist would drain him of his infinite possibilities. Surely this in part is what torture is about: that which is feared and infinitized is brought low by dismemberment, reduced to helplessness, humiliated, all of this evidently necessary because of the great perceived power of the one tortured. The greater the infinitization, the greater the perceived power of the enemy, the more must he be brought low to restore us. We must be able to reconstitute the enemy as impotent, and to tear him down from his "high horse", just as monuments have of late been toppled. All this, I suggest, for the sake of our constitution and our "seeing as".

Here, Girard is useful. According to him, persecution and violence arise during times of crisis, which are evident by the "loss of difference", a kind of lawlessness in the air. In an attempt to address the crisis a scapegoat is sought. Because the crisis is lawless and an

aberration, the victim is sought by similarity among the aberrant: the lame, deformed, and stranger. The scapegoat is both weak enough to be overcome and yet fearful and powerful enough to account for the crisis. It is guilty and becomes synonymous with the crisis itself.

> *Terrified as they [the mob] are by their own victim, they see themselves as completely passive, purely reactive, totally controlled by this scapegoat at the very moment when they rush to his attack. They think that all initiative comes from him. There is only room for a single cause [of the crisis] in their field of vision, and its triumph is absolute, it absorbs all other causality: it is the scapegoat.* (Girard 1986, p. 43)

Girard makes this interesting observation regarding the monster, the terrifying cause for the crisis:

> *A monster is an unstable hallucination that, in retrospect, crystallizes into stable forms, owing to the fact that it is remembered in a world that has* **regained stability** *[emphasis added].* (Girard 1986, p. 33)

We see here in Girard's study of the persecution and violence signs of saturation and their connection with the ensuing violence. The "loss of difference" indicates an inability to "objectify" and constitute an understandable world. They are at a loss as to how to go on. The scapegoat is identified through a kind of pairing, by the appropriate similarity with the crisis that engenders the saturated experience. Once paired, the scapegoat and the consequent violence intend to bring the chaos of disorder under control and to restore stability. It is, however, only after the "monster" has been undone, been triumphed over, that it comes into view, that it can take on the visage of an intentional object, one now "objectified".

Consider, again, the woman raped. She wants to be able to restore her previous constitution of the intentional object of her home. The rape has made her incapable of doing so. Instead, the intentional object is now invaded by the interruption of the rape and rapist. The saturation is over her, and constitutes her as someone anxious and unsure. She apperceives the excess and saturation as soon as she enters the house, but most especially in her bedroom. The criteria that previously were associated with the intentional object of a safe and secure abode now are inextricably associated with that night when he violated her space and body. With the aid of primal phenomena, it was easy to constitute this house, this space, as hers, safe and comfortable. Now the space has been contaminated by his continuous presence, always intruding into the space that seems no longer hers, or no longer only hers.

How was it that she was able to previously accomplish this house as safe and hers? Is it by its minute familiarity, every space and corner named and accounted for? Here is where there are forks and knives, here cups and dishes, here sheets and towels, here familiar clothes, shirts, pants, and shoes. If she leaves her toothpaste out in the morning, it is still there, right where she left it, when she returns home? Everything, every corner, is named and familiar. At any time or place in the house, she knows where and how to get to anything else. It is an extension of her body, or her presence. This can be said of nowhere else on earth. It is hers. Now it is violated, all of it. He lurks everywhere, not just as present, but as menacingly present. She can still easily find her toothpaste or shoes, but it is the aura that is different, that aura that had silently said "you're home and safe". Another part has been added, been constituted and synthesized from the familiar evidence of her house, the furniture, tables, chairs, and all that was familiar as hers. Now it is haunted and will not go away: what is past intrudes on the present, coming towards her now.

The rape victim fails in being able to constitute *heimlichkeit*, and she knows it. What instead takes its place we might call *unheimlichkeit*, which is primarily known by its negation of *heimlichkeit*. The *unheimlichkeit* is present, tangible, near as the stench of death, suffocating. Certainly, she resists noetically accomplishing the noema *unheimlichkeit*, whatever that can mean. It is unclear how the constitution of a negation would be accomplished, except perhaps as something missing, the constitution of intentional disappointment. Instead of the *heimlichkeit* she knew and longs for, she might try to constitute something else,

perhaps being-in-the-street, being-in-a-motel-room, or being-in-someone-else's-house. She is unable, however, to accomplish any of these in her home. What she is given by intuition will not allow it.

Can we say that the saturation is manifest most clearly in the lack of alternatives? It is not perhaps only that she cannot constitute an "object" in Marion's sense, but that she has no intentional object available. If affairs in your home become confused, perhaps because of excessive demands elsewhere, your house falls into a kind of disorder: clothes lying all around under the bed and on the floor, dishes lying unclean and sticky, dust and crumbs visible on the floor. This being an exceptional situation, you may not have a concept for it. It may seem overwhelming, chaotic, and too much. The horizon of what it means for you to have a home has been burst and you have no alternative intentional object to "contain" it. Later, perhaps, in a way reminiscent of Girard's monster, after you have calmed down and restored the house to its previous order, you come by such a conception, a noema, and you call it a "messy house". You now have an understanding of what the horizons of a "messy house" are and how it relates to external horizons, such as what cleaning it up looks like, and the possibility of restoring it to a familiar intentional object. In this sense, what began as wild and overwhelming becomes something tamed and brought into the everyday world. Is it possible that in a significant measure what saturation reflects is this lack of a sufficiently rich set of available noema? In part, this is perhaps what religion and mythology do for the numinous. They populate and give form to what is formless, murky, and not able to be "objectified".

Is it not possible that this is what revenge does: addresses the formlessness of the saturated phenomenon that is her *unheimlichkeit* and finitizes it? This is what noesis does: it gives form. It gives morphe to hyle. It bestows meaning. If Marion is right about saturated phenomena, there are some actions that play a role similar to noetic activity, but do not bestow meaning, at least not in the kind that constitute a noema. Some action is required in order for the saturated to show up at all. Something is required for the *unheimlichkeit* to disclose itself, even if it is something not definite, something ineffable, something excessive.

In the receiving of a saturated phenomenon, something similar to noetic activity must be correlated with the reception of what is given. In *Ideas I* (Section 92), Husserl speaks of the attentional ray changing, and as it does so correlative noematic–noetic modifications take place. If, as Dahl argues, saturated phenomena interrupt the everyday and are introduced through the everyday, then the everyday and familiar intentional objects must be present somehow in and perhaps prior to the reception of the saturated phenomenon. We have argued that even if during this modification of the attentional ray the object changes from one familiar to a saturated "non-object", there is noetically accomplished the noema of "failure", or of a lack of alternatives, or something like this, and that this noematic–noetic correlation is present and associated with the saturated phenomenon.

The noetic activity of cleaning up and restoring the house to its previous order is an eating away at the saturation of the indescribable and excessive "mess". It is like a pioneer or an explorer heading out into the jungle and unexplored territory, cutting a small path through the dense brush, one step at a time. This is what the revenge is like.

Perhaps we need to ask what activity, we will call it noetic, is entailed in taking revenge. We have to in some sense take hold of someone or something, with this someone or something being held somehow connected to what engenders the revenge. It is backward looking to what has been. We may be attempting in some sense to change or wipe out the past, as if it never happened. And this reminds us of Eliade's creation myths (Eliade 1959), wherein there is sought a cure for the weariness of the world by returning to the beginning: to the time of creation. Revenge, in this sense, has a strong nostalgic sense of a better past and a strong desire to return to that past now. Perhaps by wiping off the surface of this planet the existence of such a person, killing, cutting up, and incinerating, wiping out all signs that he ever existed, it would almost be as if he never did exist; and if he never did exist, then what happened in the past could not have happened. The past without him is

restored now. Can this be our paradigm, all other forms and modes of revenge deriving and related to this ideal?

If this woman who was raped could wipe from her memory that the rape ever happened, she would be free of him. Her home would be returned to her. Here, perhaps is another paradigm, much like the former.

Marion, in speaking of violence before Levinas' Face, says:

> *Why does the non-objectifcation of the idea of the infnite expressed in the face scare me to the point that I wish to defend myself from it by going all the way to the possibility or even reality of murder? Most probably because, in this case, my consciousness loses its habitual mastery by losing its power or its permission to objectify: "here the inexhaustible surplus of the infinite overflows the actuality of consciousness". Such an overflow indicates a shift or even a conversion of consciousness: before the face, one must, in order to see it, pass from theory (thus from intentionality and the constitution of objects) to ethics (counter-intentionality, election, respect of "holiness") by a dispossession of the transcendental posture of the ego, by a dispossession of possession itself. Violence is born from the refusal of this dispossession.* (Marion 2020)

Vengeance is surely a form of violence. If the aim of violence is founded on the aim of the transcendental ego to constitute "the whole of beings as objects" (Marion 2020), then this must, likewise, be the aim of vengeance. This is just what we have been arguing: that vengeance has a noetic-like character inasmuch as it aims to constitute a safe home, and the sense of *heimlichkeit*.

Of course, not every saturation engenders violence or revenge. The saturated experience does, however, tempt us to master it, to bring it under our control, so that we can have it, rather than it us. As such, it initiates our constitutional apparatus to "objectify". We have already suggested that this "objectification" prepares the way for violence, for a thematic enactment. Still, this does not account for why of all possible "objectified" responses violent revenge would be selected.

The distinction, we have suggested, is accounted for not in the saturation itself, but in the familiar. It is the context of the familiar and everyday that the saturated event is given its specific meaning. We have suggested that the distinctive might be located in the experiences of negative saturation and in harm. Negative saturation alone will not account for the impetus to revenge. Otto's numinous, while presenting as a negative saturation, does not engender revenge. Harm, while in itself a form of negative saturation, is not sufficient either. Scheler's distinctive appears helpful here. Revenge requires not only impotence, which would be encountered in any saturation, but also harm attributed to one guilty for whom one might judge oneself equal or nearly so.

The picture that emerges in revenge is that out of the familiar and everyday the alien interrupts the everyday. For Dahl, the holy is disclosed in and through the everyday. The religious response tames the alien, gives it form, without ever fully encompassing its alien and mysterious nature. We would suggest the same for revenge. It must work in a way similar to Girard's scapegoat, where the terror and impotence of a crisis is addressed through the murder of the scapegoat, ultimately relieving the spiritual crisis in ways that are not fully transparent to us. They remain in part mysterious and hidden. Indeed, it may very well be that the hiddenness is necessary for the ultimate release. We must, to the end, believe that the scapegoat is both powerful enough to have brought the crisis on and that we are powerful enough to kill the scapegoat. The scapegoat has magical properties, just as its killing is magical (Girard 1986, chp. 4).

If this is to make any sense, we must recognize that, as for the scapegoat, so too for revenge, we have entered into a mythological realm. Entrance to this mythological realm was prepared both by the everyday crisis and harm and by our noetic ability to "objectify" both the one guilty and the event, making possible the enactment of a mythological drama where themes and meanings play. We are engaged in ritual, not the dead rituals of convention or habit, but ones that work and have power.

## 11. Revenge and Myth

What "objectification" enables is the constitution of meaning. These meanings enable connections that are not necessarily possible in the "natural" world, even as they might be in the world of imagination. By metaphorical extension, concepts can be united and related that might otherwise appear disparate and unrelated. Events separated by time and space can be united into a whole, a temporally extended intentional object, and into something such as a story or narrative. Here, Good can be seen as doing battle with Evil, with its ultimate defeat. The rape of a woman is thereby elevated and lifted up into an imaginative realm. The particulars of space and time are now more than this particular woman in this particular apartment. Instead, they are enjoined to a larger thematic battle in a realm where themes can do battle, and this is an amazing thing. Battles are usually fought with flesh and blood, bruising and pain. It is an embodied engagement that fills and overwhelms our senses. They are terrifying.

It is in this uplifted imaginative realm that saturation dwells. The insult lodged against someone does not leave an apparent, visible mark. It strikes home in this invisible realm where meanings thrive and dwell. This is even the case for Othello. Outwardly, Desdemona's betrayal, had it actually occurred, would have only been evident in her absence. The connection between Othello's torment and her absence is obscure and invisible. It can only be glimpsed by entering into this mysterious realm of meanings that elaborate what is seen.

Can we even begin to understand this invisible connection between the harm done to one and the revenge sought on the one guilty without some kind of story or narrative? We have spoken of the importance of pairing for the establishment of a context for revenge to work itself out. But this mere association is not sufficient to account for the particular form of revenge. More is required. We require something such as a story. The outlines of the story have their beginning: the beloved daughter is dead. What, if anything follows from this? You bury her. You mourn her loss. Already, even here, there is a story: a temporally extended intentional object, an object that was already prepared, already given to you. You did not have to collapse in a chair, dumb like a stone, not knowing what if anything follows. No, instead, you are swept along in a way no different from constituting the presence of a tree. There is an already existing noema and a noetic activity that accomplishes the presence of the tree. The noema, or perhaps we might say gestalt, is already given. The announcement of the death of the daughter activates various noemata as a way to comprehend what is given.

Everything changes, and a new set of noema are activated, when the daughter is declared "murdered". When the daughter was "only dead", she could have fallen while rock climbing or succumbed from a lost battle with COVID-19. Murder changes everything. Now there is one responsible, one guilty. We sometimes distinguish the deaths by saying one is "accidental", the other "intentional". The revenge motif is most generally activated in the latter case and not in the former.

The daughter's murder is experienced as a saturated event. You cannot find your footing, and some might even faint. How is this possible? It makes no sense, and yet you cannot get away from it. It is hard and unyielding. No manner of turning your head this way or that is capable of bringing it into view. It waivers between conflicting intentional objects: between the daughter that was bright and alive this morning to the unresponsive weight of her graying corpse. You want to make sense of it. You want to escape this dizzying impotence.

The "objectification" of the saturated event we are suggesting enables a way forward. It not only masters the saturation, or at least begins to, but enables the lifting of the saturated phenomena into a meaningful realm where sense can be made of it.

This thematic realm bears some resemblance to a Schützian province of meaning, something like fantasy or art. It cannot be exactly like these, however, because their "cognitive style" lacks the kind of play and "tension of consciousness" that a revenge drama would entail. Barber speaks of the "pragmatic" character of Schütz's working world.

He says, "In the pragmatic mode mastery is chiefly at issue" (Barber 2017). Effort in this pragmatic mode is focused towards the achievement of a specific goal, which is why its "tension of consciousness" is high. Here, we are dealing with a "real" world where actions or inactions have important consequences, which is why our attention is active and focused. Non-pragmatic provinces of meaning, such as art, theater, or humor, lower the "tension of consciousness" by opening us up to aspects of a situation that the demands of the working world would not allow for the sake of a project. As such, the urgency and seriousness of revenge possess aspects of the working or pragmatic world. It even shares a similarity with what Barber calls "hyper-mastery", where all other provinces of meaning are excluded in a kind of absolutization of the pragmatic (Barber 2017, Section 2.5). In revenge, we become consumed by the project. But, on the other hand, revenge is not like work, where we are invested in feeding or sheltering our families. He says that of the pragmatic province that it "is based on an *ego agens* gearing into the world and exercising mastery with confidence based on the idealization that 'I can do it again'"(Barber 2017, p. 35). Pragmatic practices are generally regularized and repetitive. It is in the regularization that mastery is achieved. We have learned how to sow seed to produce in most years an abundant crop. Even while revenge may seek mastery, we do not feel like we have mastery over the situation. If anything, it, like other forms of saturation, feels like it has mastery over us.

It is perhaps in the Schützian epoché that revenge is most unlike the working world. In the pragmatic–working world, we suspend judgments regarding the reality or existence of entities (Barber 2017, Section 2.5). Essentially, in this working world we adopt non-judgmentally the natural attitude. However, we have been essentially arguing that in revenge the natural attitude is abandoned. We do not believe that revenge can be best understood as some kind of calculative, pragmatic enterprise, where we calculate that the revenge will achieve some future pragmatic goal.

Instead, the province of meaning associated with revenge bears significant characteristics with Eliade's understanding of religion as given in his *Cosmos and History*. He says,

> In the particulars of his conscious behavior, the 'primitive,' the archaic man, acknowledges no act which has not been previously posited and lived by someone else, some other being who was not a man. What he does has been done before. His life is the ceaseless repetition of gestures initiated by others. (Eliade 1959, p. 5)

> The crude product of nature, the object fashioned by the industry of man, acquire their reality, their identity, only to the extent of their participation in a transcendent reality. The gesture acquires meaning, reality, solely to the extent to which it repeats a primordial act. (Eliade 1959, p. 5)

It is in light of this relationship with a transcendent reality that rituals arise. Time is regenerated by a re-enactment of the creation myth.

> For the cosmos and man are regenerated ceaselessly by all kind of means, the past destroyed, evils and sins are eliminated, etc. Differing in their formulas, all these instruments of regeneration tend toward the same end: to annul past time, to abolish history by a continuous return in illo tempore [in a time before recorded history], by the repetition of the cosmogonic act. (Eliade 1959, p. 81)

While such behavior may have vital pragmatic ends, they are not "geared into the world" in the way that we would normally associate with the working world. They are intimately and necessarily founded in a world outside this world. As such, they operate with a logic that is not necessarily bound by this natural world.

The temporal structure of the working world is radically different from that in Eliade's province of meaning. In the working world, the temporal sequence is inconvertibly ordered according to a past, present, and future. What is past cannot be undone. It is inaccessible, whereas in Eliade's province, the past can be undone. Time has a cyclical nature, and archaic man seeks to overcome the "terror of history" (Eliade 1959, chp. 4).

There is much similarity between Eliade's archaic and the one seeking revenge. Temporality in Eliade's archaic bears more resemblance to Schütz's fantasy or art. Here, temporality need not be confined to a "natural" ordering of past, present, and future. However, even as the presentation may mix the orderings, for example, what has gone after being presented before what comes before, we nonetheless, upon hearing or reading the story, remix the presentation into a chronological whole that is consistent with ordinary time. It is, nonetheless, possible with fantasy to not merely mix the order of a chronologically ordered story; one can make the future reach back into the past and affect times that come before.

We have previously spoken of the importance of pairing in order to understand how the relationship between the one harmed and the guilty one is established, thereby allowing an imaginative play of meanings that constitutes revenge. We can find something very similar in the study of magic.

In Sir James Frazer's classic work, *The Golden Bough*, he says of the Principles of Magic:

> *If we analyse [sic] the principles of thought on which magic is based, they will probably be found to resolve themselves into two: first, that like produces like, or that an effect resembles its cause; and, second, that things which have once been in contact with each other continue to act on each other at a distance after the physical contact has been severed. The former principle may be called the Law of Similarity, the latter the Law of Contact or Contagion.* (Frazer 1993, p. 11)

He goes on to say these two principles are essentially applications of the "association of ideas" (Frazer 1993, p. 12). Frazer continues in this vein detailing how these principles are employed to injure an enemy, facilitate childbirth, to heal and prevent sickness, and secure an abundant supply of food (Frazer 1993, pp. 12–17). The form that the magic takes is suggested by similarity with the aim of the magic. A medicine man will go through similar contortions that a sick person is experiencing, subsequently being "healed" of the sickness. The closeness of ideas through similarity is conceived as being close to what the ideas refer to.

Such principles are not merely found among "primitive" people. We can also find them in alchemy in trying to transmute "baser" metals into gold. Lawrence M. Principe, in his work on *The Secrets of Alchemy*, speaking of the making of the Philosopher's Stone, through which "baser" metals can be turned into gold, says that "the prepared substance or mixture is placed in a glass vessel with an oval body and a long neck, often called a philosophical egg on account of both the size and the shape of the belly, and its function in "giving birth" . . . to the stone" (Principe 2013, p. 123). This material is then heated for some time until it turns black, indicating the "death" and putrefaction of the substance (Principe 2013, p. 124). After more heating, the stone will turn white, indicating that it is capable of transmuting all metals into silver (Principe 2013, p. 124).

We see here an application of the Principle of Similarity, whereby the belly of the heating flask is similar to that of a pregnant woman; the blackness indicates a likeness to the death of life, and the white stone is similar to the white of silver. The Philosopher's Stone was supposed to work like leaven in kneaded bread or a seed that sprouts according to its nature (Principe 2013, p. 126). The alchemists see similarities with other natural processes and reason that such similarities have application beyond their particular contexts.

We can see the same in modern science. Galileo studied balls rolling down an inclined plane. From his studies of a single plane with a few balls, he reasoned by similarity with all inclined planes, even to the general nature of the attraction of all bodies to the earth, from which Newton developed his general theory of gravitation.

How, then, is magic different from modern science? To answer this question would require far more space than can be allotted to it here. Our chief point is to motivate the notion that Husserl's notion of pairing and the principles of magic are omnipresent. What is more, it is plausible that their application may be less "judicious" than modern "rationalism" will allow.

We have, then, enough to imagine some kind of understanding of the Schützian province of meaning. We require, however, more details. It is, we are arguing, some kind

of mythical realm, but what exactly are the details of this mythical story? Having lifted the context of revenge into a mythical realm where narratives and rituals are at play and make sense, we would want to give some account for why the story or ritual takes this form and not some other. What troubles such a project is the sense that in this imaginative and fantastic realm, it is themes and concepts that rule the day, and there are many such themes possible. And yet, if we were to survey the kinds of myths and stories that mankind has told itself, the list is not overly long. That men and women are capable of such story telling is indisputable. What is less clear is not only the uses we put to them, but why and how they bind us. They make the world we live and dwell in, preparing it for us before we are there. So too is this the case for the revenge story and its associated ritual.

Revenge is characterized by an impotence before near equals. As such, it bears much similarity to Girard's scapegoat phenomena. Here, the tension and anxiety engendered by a crisis are addressed by killing the scapegoat that stands in for the crisis, subsequently relieving the tension and anxiety. The scapegoat must be regarded as sufficiently terrifying to actually be the cause of the crisis and yet weak enough that the mob can believe that they can actually kill it. As Girard tells it, the scapegoat is not a formalized, symbolic killing. The mob actually and necessarily believes that the scapegoat is responsible for their woes. Without such belief, the killing would not actually relieve tensions and anxieties. The same is true for all acts of magic. We want to say that the same is happening here in revenge.

The harm of revenge is insurmountable. It is associated with a past event that has temporal legs, its shadow cast upon our temporally extended experience. Before this harm we are utterly impotent, not unlike the crisis of the scapegoat. Revenge's most specific harm cannot be seen. It is psychological; we might say spiritual, matters that are difficult to get at. Because such things as grief, remorse, guilt, and resentment dwell in an imaginative, meaningful realm, it is not surprising that their remedy might be sought there as well. It is only in such a realm that tit could sensibly be traded for tat, where the rapist must be raped, the betrayer betrayed, and the murderer murdered. The familiar establishes the context for the negative saturation which is to be healed. The similarity of the tit and the tat is sufficient for the magic to work, where your rape of me is somehow healed by my rape of you. We say that we are even, but this "evenness" can only occur in some imaginary, metaphorical realm. What happens here is similar to what occurs with the scapegoat. The crisis continues even after the scapegoat is killed. Despite the "exchange", I remain still raped and my daughter still dead, and yet in some sense we are better off.

How we come to anticipate the satisfaction of killing the scapegoat or the accomplished revenge is unclear. Perhaps it is given to us as a cultural motif, or perhaps it runs deeper.

If we return to impulsive acts, perhaps we can find a hint. These, as you recall, Scheler rejects as acts of revenge because they are not delayed. If a mosquito tries to bite us, we "instinctively" take a swipe at it. The same is the case for Scheler's "box on the ear". On the other hand, if a steer knocks you down, your "instinct" is not to take a swipe at it. Rather it is to flee. This "instinct", perhaps biologically derived, reflects something of the structure of revenge. We do not seek revenge on those for whom we regard as superior in some sense but only upon those who we regard as equals. It is plausible, then, to consider this "instinctive" trait to be what connects the tit with the tat. Even if this is the case, we find in revenge a symbolic and imaginative elaboration of the possibly originating impulse, the very kind that relies upon "magic" and pairing. As we have already indicated, harm has been symbolically elaborated from "mere" physical harm, so that the saturated experience is analogous to physical pain. It is plausible, then, that the impulsive response has now been replaced by revenge in response to the pain of saturation, the one guilty replacing the mosquito.

If this is correct, what can we say of "instinct"? We have to believe that it is associated with our survival. If so, we must regard the one guilty as a threat to our survival, but not in the same way that a mosquito is. One in the violent act of revenge, harming another, feels as if it is similar to impulsive acts of frustration. If so, then it is as if the guilty one is resisting you and in your way. There seems to be some sense in which the guilty one is in

the way of you noetically accomplishing some intentional object, whether it be a safe home, a faithful marriage, or a living daughter. It is perhaps as if the revenge is trying to clear away the impediment to that preferred state.

If this is right, what must first be determined in the trajectory of revenge is what is perceived as crucial in the negative saturation, whether it be the loss of pride, of self-image, of impotence, or insecurity. What drives the desire for "justice" is the negative saturation of disorder, of lawlessness, of fear. By controlling, capturing, or corralling the guilty one, the saturation is eliminated, and peace and order re-established, not unlike the murder of the scapegoat. So too for Othello, as it is his understanding of himself that is overthrown and sends him into a tizzy. By the murder of Desdemona, his sense of self, so he thinks, his pride, his manhood, are re-established, and so too for the daughter murdered. It is not so much about the daughter as it is the sense of helplessness gone mad, a helplessness remedied by the violent murder of the one responsible. This does not really address the grief that follows, but it aims instead at the helplessness. The aim of revenge is to rid oneself of that pesky "mosquito" imaginatively engendered, the negative saturation, and this is accomplished by a magic we believe in, that is, by real magic.

## 12. Murdered Daughter Revisited

Your daughter is dead. You had anticipated seeing her that night after helping on New Years' Eve to transport safely home those who had had too many. Instead, you are told that she was in a fatal car accident, killing her and injuring others. You are undone. You cannot find your footing. This is the saturated event, but it is negatively saturated because of what you can grasp: that your beautiful, vibrant daughter is dead, and the shock of a disappointed anticipation.

Later you learn that it was a drunken driver who was the efficient cause of the accident that resulted in her death. Now is added another entity to the complex experience. There are now three, your daughter, this man, and yourself, now paired in a complex "object". To the negative saturated experience of your daughter's death is joined the drunken driver. This drunken driver is readily constituted as an intentional object. You understand what it is to be drunk. You constitute him as reckless, self-consumed, perhaps out of control. We now have the saturated death of your daughter and an unsaturated constituted drunken murderer. The saturated death of your daughter that constitutes you is now paired with the definitive, demarcated, intentional object of the drunken murderer, whom you now constitute, resulting in a diminished saturation, a growing sense that you are no longer impotent.

This is a new kind of "object:" this mixture of saturation and unsaturation. Of course, every saturated phenomenon contains a mixture of such. In the phenomenon of the holy, Dahl indicates that it occurs at the interface of the everyday and the alien, at the interface of familiar objects and those "objects" that resist "objectification". Something similar is going on here in the case of the daughter's murder.

At the interface of the everyday and the alien, there are two broad ways of proceeding. Dahl refers to these as "continuity" and "discontinuity" (Dahl 2010, chp. 7). In continuity, the everyday is "elevated" to the holy (Dahl 2010, p. 267), that is, for the everyday, the familiar, the "objectified" to "objectify" the non-object of the alien saturation, and draw it into the familiar. In this sense, it is a "reaffirmation of the everyday" Dahl 2010, p. 271). Speaking of worship as a discontinuous response to the holy, Dahl says

> [W]orship can only call attention to the holy and bring it to the fore in so far as it is capable of being something like a dramatized phenomenological reduction that frees the holy from the downward thrust of the everyday. Consequently, worship must pose as an interruptive event by marking out its discontinuity with the everyday. (Dahl 2010, p. 271)

Dahl here sees "discontinuous" worship as a Schützian transition from the everyday province of meaning into a religious sphere. The question for us is which of these two responses to the interruption of the everyday occurs in the case of revenge.

Dahl refers to the "discontinuous" response as representative of "worship". Worship has a positive valence relative to the holy and alien, and appears most appropriate to a "positive saturation". "Negative saturation", like Otto's daemonic–divine, results more in dread, Otto avers, than in fascination. As a result, one seeks to flee the experience. As such, it seems that in such cases, Dahl's "continuous" response is most likely pursued, whereby one attempts to drag the alien, the source of the saturation, back into the familiar. Inasmuch as revenge is a response to negative saturation, we would expect the same for revenge.

This analysis suggests that in revenge, the one harmed, the one caught up in the negatively saturated event, seeks to respond by de-saturating the experience; and that is accomplished by way of the familiar, by "objectifying" the non-object. This is accomplished by turning from the murdered daughter towards the murderer. The dead daughter elicits a different kind of saturation: that of grief. To that saturation you do not flee. Instead, you dwell in it, but the time for it has not yet arrived. In revenge, it is the guilty one that stands out. Him you can "kill" with objectification; him you can abstract and steal his flesh and blood in mere ideas and meanings.

In this way, the guilty one is lifted up out of his particularity into an impersonal realm of meanings, where Frazer's principle of association can play. This is a different kind of Schützian province than one of religion, at least as Barber has it. The tension of consciousness in Barber's understanding of the religious province is relaxed relative to the working world. Barber says,

> The finite province of religious meaning releases the ego agens from its restricted focus on its own perspective by locating it within the broader perspective of a friendly ideal power which shares responsibility for the details of the ego's history and/or cosmological under-pinnings and whose long term governance produces personal and communal liberation. (Barber 2017, p. 88)

Clearly, this does not describe the mixed realm of the familiar and alien associated with revenge. Here, the province is not inhabited by a "friendly ideal power", but rather by something malevolent and evil. However, bearing in mind Scheler's prescription for revenge, this malevolence is not understood to be all powerful, but rather one that is a near equal, one upon whom revenge and harm could be inflicted. In this we see something of the noema according to which the guilty one is constituted by the one harmed.

The genesis of the guilty noema is beyond the scope of our investigations. We suggest, however, that it might very well vary from tradition to tradition. It is an understanding that we somehow inherit. With this inheritance is not only the noema of the guilty one, but also of the avenger, and, in this case, the daughter. All have been paired, all in some sense "objectified" and constituted as intentional objects that play a role in the "objectified" drama, all for the sake of undoing, diminishing, or crippling the negative saturation.

The violent revenge is not so much about the daughter as it is the sense of helplessness gone mad, the negative saturation that overwhelms, a helplessness remedied by the violent murder of the one responsible. This does not really address the grief that follows, but it aims instead at the helplessness, and the restoration of something more familiar, something less drenched with the alien and saturated.

## 13. The Dialectic of Revenge

We have one last suggestion to explicate the structure of revenge. Consider the situation of the rape previously considered. In this case, the woman has paired the past intentional objects of the home, herself, and the present intentional objects after the rape. The post-rape intentional objects haunt the older understanding. She cannot synthesize the two into a stable intentional object.

Why is not it enough to simply say, "I was raped"? It is in the past and no longer now. In this sense, it is out of reach, in a sense no longer mine. This is troubled by our enduring selves that unite temporal moments into this single "mine". Moreover, the apartment itself serves as confirmatory evidence for both the past *heimlichkeit* and the present *unheimlichkeit*.

The two are enduring temporal intentional objects. So a conflicting and contradictory appresentation endures.

It seems that attempting to somehow resolve the contradictory is incomprehensible. We cannot cohere them. One could imagine here something like a Hegelian dialectic whereby the seeming contradictions are probed in an attempt to synthesize them into a whole. The encounter with this dialectical process is experienced as saturated because no single concept can be constituted.

We have, then, these "premises": *heimlichkeit* and *unheimlichkeit*. Revenge seeks to restore the *heimlichkeit*, and therefore eliminate the *unheimlichkeit*. But it is likely impossible to "fully" restore the original *heimlichkeit*. The hope of revenge is to "soften" the tension. The *unheimlichkeit* is "infinite" and saturated. By finitizing the rapist, the *unheimlichkeit* is conceptualized and dragged back into the familiar. He is now a convicted criminal, no longer the untamed and anonymous rapist. The *heimlichkeit* can be perhaps re-established as "safer". No longer is it "naïve", but now a nuanced, and transformed *heimlichkeit*, where certain particular and familiar measures can be taken to secure or more secure the *heimlichkeit*. In this process, the "naïve" *heimlichkeit* has been probed and aufgehoben. But it seems that the same has been accomplished for the *unheimlichkeit* by being finitized. The new *heimlichkeit* incorporates aspects of both the older *heimlichkeit* and the older threat and insecurity of the saturated *unheimlichkeit*. This is what makes the synthesis not naïve.

But in order to accomplish this, two saturations had to be dealt with. The first is the saturation of the dialectical moment, and the second the saturation of the rape itself, its *unheimlichkeit*. Revenge engages the dialectical moment. Having not done so would have been more like the response of ressentiment, where revenge is never taken. Likewise, it accomplishes the second by the act of the accomplished revenge itself, whether by violence or imprisonment.

## 14. Conclusions

The structure of revenge consists of one harmed, the perception of harm and its associated suffering, and one perceived as responsible for the harm. In the case of revenge, the situation is apperceived as a negatively saturated experience. As such, the one harmed feels "out of sorts", and "overwhelmed". As distinct from a positively saturated experience, one senses no or little sense of allure. Instead, as Otto says, one desires to flee. But how is one to flee? The negatively saturated experience binds and has a hold on you, constituting you as enthralled. Revenge seeks to remedy the situation by the intentional act of objectifying, constituting, and finitizing the infinite situation. This is accomplished by constituting the guilty one as guilty. Already in doing so the situation becomes more understandable and familiar. The slave who, as Scheler has it, would never seek revenge on his master will likewise never constitute his master as guilty. It is, however, in the accomplishment of the revenge that the guilty one is finitized. He who we can harm must be finite. In this way, the saturation does not have you. You, in some measure, master it.

We have suggested that the realm and machinery required for this process arefound in the realm of the imagination, where similarity and association of ideas and concepts are at play. Saturation plays at the edge of this realm as alien. It is by way of the familiar and constituted that the alien is tamed and revenge puts the situation to "rest".

In drawing distinctions between various levels of phenomenological analysis, Giorki's distinction between "phenomenological psychological reduction" and "transcendental reduction" can prove useful. Giorki et al. say of the latter that it "shifts the analysis to another level of consciousness beyond the psychological. One could call this a philosophical level of reflection whereby the researcher becomes aware of the *conditions* for the possibility of any experience" (Giorgi et al. 2017).

They go on to say:

> The transcendental reduction aims for a completely purified consciousness which has no relationship with anything empirical. The phenomenological psychological reduction does not achieve that degree of purity.

> *Transcendental reduction speaks about any possible consciousness but no real conscious-*
> *ness. But as psychologists we are interested specifically in actual human consciousness so*
> *we do not bracket the positing of consciousness itself.*
>
> *Because of the assumption of the psychological attitude toward the data, the essences that*
> *are apprehended are psychological essences and not philosophical ones. Psychological*
> *essences are typical, not universal.* (Giorgi et al. 2017)

Giorgi et al. regard the transcendental reduction as in some sense "deeper" than the psychological reduction. The steps followed in the transcendental reduction are similar in the psychological reduction. The difference is largely reflected in the level of the reduction. There is still an emphasis on description and with an eidetic reduction, but the level of the reduction is different. It does not aim for a "pure consciousness", but nonetheless aims for conscious contents, the likes of fear, shame, etc.

In exploring revenge, we have passed through many "levels" of analysis: eidetic analysis of the natural attitude, the psychological phenomenological level, and employing as well fruits of transcendental analysis. In doing so, we attend to varying aspects of the phenomenon. At the "sociological" level, we may gather data relative to the phenomenon of revenge: instances where people say they are seeking revenge, what they do and what they say. We might say that this is an analysis of the natural attitude. Scheler speaks of a "tit for tat", one harm for another. Going deeper, we might ask what exactly is "harm". How do we recognize it? Indeed, if we are to "harm" another in return for their "harm", we need to be able to know what constitutes "harm". This represents an eidetic analysis of the concept. This "harm" is, nonetheless, not specific to the "harm" associated with revenge. What distinguishes this "harm" from other "harms"? Here, we are exploring an eidetic analysis of "revenge". The "harm" we find must be associated with another whom we judge to be guilty. This discovery discloses that some sort of pairing is established between the harm, the one harmed, and the guilty one. Pairing is a higher-order capability of human consciousness, one revealed by a "higher" order attending to human consciousness itself, often associated with a transcendental reduction. Even so, the puzzle still remains, despite the unity of a complex intentional object, why one harm would lead to another. Again, from the transcendental analysis of human consciousness, derivative from constitution of intentional objects, we explore saturation, an aberrant constitution, and explore its application to the particulars of revenge.

We have explored here revenge almost exclusively from the perspective of the one seeking revenge. A fuller exploration would include the guilty one. The avenger and the guilty one form a unity. The former understands the guilty one in a certain way, and this informs, even enables, his or her revenge. Different understandings of the guilty one would seemingly result in different responses, in the activation of different complex intentional objects, within which the two interact and play. One such possibility is forgiveness. In addition, the phenomenology of the guilty one as guilty has not been explored at all, and this would be important in understanding the two sides of forgiveness: from the perspective of the one harmed and from that of the one doing the harming. Forgiveness, genuine forgiveness, regards the suffering differently than revenge. Rather than fleeing it and seeking to overcome the saturation, it engages and remains in it.

**Funding:** This research received no external funding.

**Conflicts of Interest:** The author declares no conflict of interest.

## Notes

[1]    A "?" has been removed at the end of the cited text. It looks to be clearly an error.

[2]    Otto argues, however, that the experience of the numinous is inherent in human "psychical nature (Otto 1936, p. 128). It would seem that something similar must be said of the experience of saturated phenomena: humans are already in some sense inherently capable of having such experiences. The capability is not a social construction.

3    The event referenced here and throughout actually happened in Los Alamos, NM, on New Year's Day 1988. Jack Kerns, 31, was driving a pickup. He had been drinking too much and swerved to avoid a parked car, crashing into a car driven by Jennifer Fleming, 16, a LA High School junior. The crash killed her and seriously injured the three other occupants of the car. Jennifer was, ironically, providing transportation to those who were too impaired to drive home safely. I got to know Jack Kerns in the Santa Fe prison. A scholarship was established by Fleming's parents in her name. As of at least 2012, it was still in effect. Jack Kerns desperately wanted to speak to Fleming's parents, to tell them he was very sorry, but they refused any contact with him.

4    Martin Heidegger in Section 31 of *Being and Time* speaks of understanding as "being able to manage something." We take this as being able to know what one is about and how to get about.

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
