# Peer review of "The Puzzle of Revenge"

_religions, doi:10.3390/rel13050444_

Round 1
Reviewer 1 Report
I found this author's analysis to be compelling and thorough, and I think it makes a contribution to the literature on the nature of violence. I think it is publishable, as is. I do have a few comments that the author might find useful either for this piece or in future works.
1. Is 'missingness' (ugly word) the same as loss? It seems to me that they are different experiences with different intentionalities. Missingness is accompanied by a horizon of expectations which may be extended without end, or which may result in finding, or loss. Missing persons are good examples. Without further information, relatives of the missing often hold onto hope, indefinitely. Sometimes, their hopefulness is justified and the missing person is returned. Sometimes, not. It the latter case, the response may be what we can call 'closure', or perhaps it results in the desire for revenge. Loss is the experience of missingness without hope. Especially that this particular person can never be replaced.
2. There is a way in which the victims of sickness enact a kind of revenge against a disease. Cancer victims are sometime told to get angry at the disease, not to succumb to it. This involves an imaginary objectification of it as something with evil intent, as a kind of malicious subject. So anger may be a response to a disease, certainly, and their may also be a response that treats the disease as if it can be personified. However, those who have been victimized by disease (relatives, friends, or the hosts of the disease) sometimes set themselves on a further course to defeat it or vanquish it. (I have known a few doctors, who became doctors just for this purpose.) I would argue that the emotions and motivations associated with success on this course of action are not unlike the emotions and motivations associated with the accomplishment of revenge (at least for some).
3. Revenge seeks a particular symmetry in pain in the perpetrator and those victimized. Think about movies where the villain has committed dastardly crimes. Typically, the viewer finds the expeditious offing of the villain dissatisfying. The viewer typically wishes to make the villain suffer, but especially to suffer with the recognition that one of the victimized is responsible for it and that the villain is impotent to stop it. (The tables have been turned.) The only time a quick "offing" is satisfactory is when the villain is so powerful and amoral that to allow him to live would certainly entail further loss and/or when the villain himself is nihilistic--completely beyond care for his own life or suffering.
4. The author's analysis has obvious implications for a phenomenology of restorative and retributive justice. Both, I think, are connected to the notion of symmetry. Restorative justice assumes that equity has been achieved when things have been replaced and suffering can be monetized. However, restorative justice in this sense may always--in a sense--be considered imperfect for the victimized subject. After any violation, things can never go back to being exactly as they were. It is with this recognition that the desire for revenge is often engendered. Retributive justice is the answer to the dissatisfaction that comes with restorative justice. Much of the discourse about equity in the U.S. today is (surreptitiously) more about the latter than the former.
5. The role of imagination seems to move to the background in this piece, though it is promised in the abstract. Either it should be highlighted--the reader should be reminded when the author is discussing it--or it should be removed from the abstract.
Reviewer 2 Report
Thank you for your hard work on this. Please find my comments in the attached file. With best wishes,

Author Response
I very much appreciate your careful reading of my manuscript. I agree that I will often place comments in the text that I do not develop or pursue. Mostly I do this for my own benefit, but you are right to point them out.
I am still reviewing your comments, but there was in particular one that I wanted to highlight, perhaps in the hope of clarifying the issue for me and any readers.
In your comment relative to "section 2," you wonder about how a saturated phenomena might be felt experimentially. I had hoped that I had covered that in part. I used Marion's descriptions, as well as Otto's. For me the more important aspect of this description has to do with my linking Heidegger's understanding of "understanding" as knowing what one is about. I've thought some more about this and still think it's helpful, esp. in the context of revenge.
You also suggest that a saturated experience might be experienced as a kind of "timeless time." I hadn't thought much about this. What your comment suggests to me is that I need to say more about the encounter with saturation. There will be, it seems, a passage with a number of phases. Since I think saturation makes no sense save in the context of the familiar (and the temporal), one must always begin in the everyday. As one passes (quickly or not) into the saturation, one recedes from the familiar, but the familiar must never be wholly forgotten or lost, doing so (I argue) would result in a nothingness, some kind of characterless saturation, which is neither negative nor positive. In this passage, there is perhaps a confusion of time. It isn't so much that one enters into a "timeless time" as into a time in which one doesn't know what time it is, and this because as objects become less distinct, one's footing is lost, then one doesn't know what to expect; one can't order the future because one can't anticipate it. But I don't think this is a "timeless time." It is possible were one to become completely submerged into the saturation, and any connection with the familiar lost, that something closer to what you suggest is possible.
In any case, your comment has made me think that I will add a short section saying something of the "transition." The important thing about negative saturation is that one desires to flee it. That's how you know it is negative. As such, one does not desire to fall into it. It is the intense desire to flee it that motivates the revenge. As such, for revenge we are always playing at the surface, so to speak, of the saturation. One is not attracted to it. So, one never draws close to this "deep saturation." But perhaps I am wrong about that. I'll think some more about it.
I'd appreciate any comments you might have in this regard.

Reviewer 3 Report
The Author discusses revenge from the perspective of the one
seeking revenge and in the framework of structural phenomenology. If he or she agrees, I would suggest to discuss briefly how the topic could be dealt with if we take into account the concept of "worst violence" (Lawlor).
Round 2
Reviewer 2 Report
Thank you very much for your efforts at revising the text. I also appreciate your thoughtful response to my comment on 'timeless time'; what I meant by that is that I think an important part of the experiential side to saturation is the sense one has of being so fully immersed in it that the awareness of time seems to slip or be absent: a kind of 'in the flow' feeling, as it were. The familiar of course is not lost, rather it becomes even more present than is usually typical. As for the self, precisely the dualism embedded in common usage of these terms is part of the problem, and this is a strong legacy of both 'soul' type thinking but also Descartes. Hence I suggest not using everyday parlance to try and achieve a shift in perspective towards a more unified view (unless, of course, one is a dualist).